# The Search for the Elixir of Life: On the Therapeutic Potential of Alkaline Reduced Water in Metabolic Syndromes

Felippe Steven Louis G. Delos Reyes [1], Adrian Carlo C. Mamaril [1], Trisha Joy P. Matias [1], Mary Kathleen V. Tronco [1], Gabriel R. Samson [1], Nyczl D. Javier [1], Ailyn Fadriquela [2], Jayson M. Antonio [1,3] and Ma Easter Joy V. Sajo [1,*]

1  Department of Biology, College of Science, University of the Philippines Baguio, Baguio 2600, Philippines; fgdelosreyes@up.edu.ph (F.S.L.G.D.R.); acmamaril2@up.edu.ph (A.C.C.M.); tpmatias@up.edu.ph (T.J.P.M.); mvtronco@up.edu.ph (M.K.V.T.); grsamson@up.edu.ph (G.R.S.); ndjavier@up.edu.ph (N.D.J.); fyeflowright03@gmail.com (J.M.A.)

2  Department of Laboratory Medicine, Wonju College of Medicine, Yonsei University, Wonju 26426, Korea; ailyn@yonsei.ac.kr

3  Department of Microbiology, Wonju College of Medicine, Yonsei University, Wonju 26426, Korea

*  Correspondence: mvsajo@up.edu.ph; Tel.: +63-995-637-5313

**Abstract:** Our body composition is enormously influenced by our lifestyle choices, which affect our health and longevity. Nutrition and physical activities both impact overall metabolic condition, thus, a positive energy balance causes oxidative stress and inflammation, hastening the development of metabolic syndrome. With this knowledge, boosting endogenous and exogenous antioxidants has emerged as a therapeutic strategy for combating metabolic disorders. One of the promising therapeutic inventions is the use of alkaline reduced water (ARW). Aside from its hydrating and non-caloric properties, ARW has demonstrated strong antioxidant and anti-inflammatory properties that can help stabilize physiologic turmoil caused by oxidative stress and inflammation. This review article is a synthesis of studies where we elaborate on the intra- and extracellular effects of drinking ARW, and relate these to the pathophysiology of common metabolic disorders, such as obesity, diabetes mellitus, non-alcoholic fatty liver disease, and some cancers. Highlighting the health-promoting benefits of ARW, we also emphasize the importance of maintaining a healthy lifestyle by incorporating exercise and practicing a balanced diet as forms of habit.

**Keywords:** metabolic syndrome; alkaline ionized water; alkaline-ionized water; alkaline reduced water; obesity; diabetes; cancer; non-alcoholic fatty liver disease; microbiota; exercise

## 1. Introduction

Lifestyle diseases are one of the major causes of death globally. Known as "non-communicable diseases" (NCDs), lifestyle diseases contribute to about 71% of human mortality across the globe, with roughly 41 million people killed yearly, as per the World Health Organization (WHO) statistics as of April 2021 [1]. Diseases under this category are prolific in low to middle-income nations, accounting for 77% of the total recorded deaths attributed to people within the ages of 30–69. The major types of these diseases are: cardiovascular diseases, accounting for 17.9 million deaths yearly; diabetes mellitus; cancers; and chronic respiratory diseases [2]. Although daunting, the mentioned diseases all have risk factors modifiable through lifestyle changes, including maintaining a healthy weight and diet coupled with exercise, decreasing excessive alcohol consumption, and adherence to not smoking [3].

These lifestyle diseases are chronic, and determined by several factors. These factors may be uncontrollable, like genetics and physiology, or controllable, such as a person's environment and behaviors [4]. A person's socioeconomic status and immediate environment affect the type of diet they consume, if they can do sufficient exercise, their behavior to-

wards smoking, and the extent of their stress, which would modify a person's susceptibility towards lifestyle diseases (Figure 1).

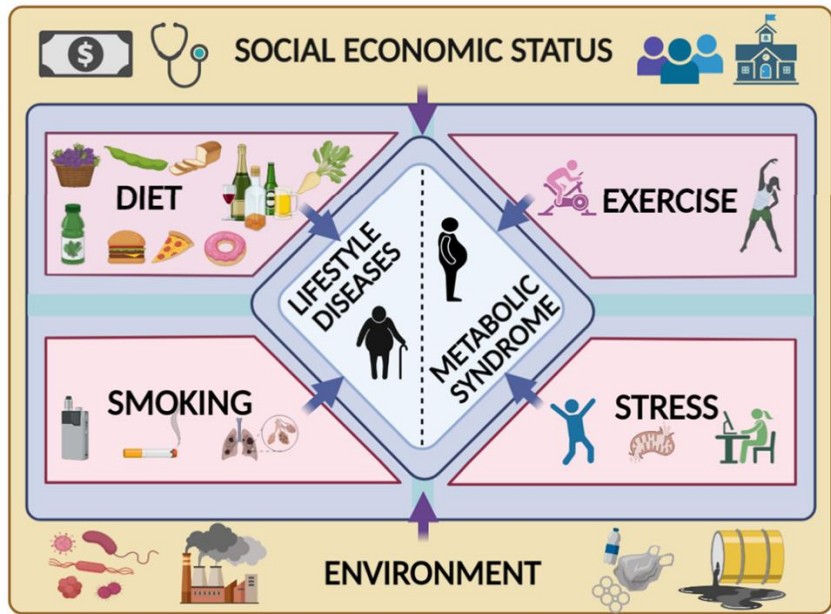

**Figure 1.** Diagrammatic representation of various modifiable risk factors that can affect the risk of lifestyle diseases. (Created with BioRender.com).

The highlighted sources of these lifestyle diseases include a sedentary lifestyle, poor quality of diet, and a stressful environment [5], which majorly contribute to the development of cardiometabolic syndrome, characterized by an aggregate of metabolic dysfunctions which situates people at a high risk of cardiovascular morbidity and mortality [6]. Environmental exposure is a significant risk factor for the development of metabolic syndrome, as observed in a longitudinal study [7]. A poor diet consisting of high-salt, high-fat, and highly sweetened food intake increases the risk of obesity and the cardiometabolic NCD index of a person. Conversely, a diet consisting of fruits, vegetables, and a healthy snack pattern contributes to lower obesity and cardiometabolic risks [5]. A more active and involved lifestyle for adults aged 18–65 is recommended by the American College of Sports Medicine and the American Heart Association to build and maintain cardiovascular endurance and muscular strength to promote and maintain health [8]. Another high-risk factor of lifestyle diseases is the continued tobacco use of the population. Tobacco use not only poses a health hazard to the person directly smoking it, but also to the people within the environment of said smoker through environmental tobacco smoke (ETS) [9]. Consequently, ETS may aggravate the risk of developing lung cancer, heart disease, and respiratory diseases in adults and young children [9]. More so, chronic smoking can be a product of nicotine addiction, making it even harder for smokers to quit entirely, since persistent nicotine consumption leads to nicotine hunger, withdrawal symptoms, and inhibited self-control [10], traits that serve as excellent lifestyle change deterrents. Effective and consistent interventions that promote lifestyle changes to lessen the risk factors of lifestyle diseases greatly help the person involved and the people in their lives.

One of the significant contributors to lifestyle diseases is metabolic syndrome. This condition or group of risk factors increases a person's susceptibility to acquiring heart diseases and other health concerns, such as stroke and diabetes [11]. The measurable risk factors for metabolic syndrome are based on the normal levels of triglyceride, cholesterol, blood pressure, and blood sugar in an individual [12]. An individual has a high probability of being at risk for metabolic syndrome if they exhibit: (1) abdominal obesity or a large waistline, characterized by a high amount of excess abdominal fat, which increases the susceptibility to heart disease; (2) higher than normal triglyceride levels (150 mg/dL);

(3) low levels of HDL (high-density lipoprotein) cholesterol, which is responsible for removing cholesterol deposits from the arteries, and then transferring it to the liver to be metabolized [13]—low HDL levels denote a higher risk of heart disease; (4) higher than normal blood pressure ($\geq$130/85 mmHg); and (5) an impaired fasting glucose or a high fasting blood sugar, which imply either an early sign of diabetes, or diabetes itself [14]. To prevent the mentioned risk factors from manifesting, besides habit changes, pharmacological and therapeutical approaches are now also utilized.

In lieu of committing to a disciplined dietary regimen and physical activities, others have already resorted to medication as an effective gatekeeper to metabolic disorders. This is done primarily to those who have shown an inadequate change in their status, even after administering preventive measures, such as exercise and a restricted diet [15]. Given the interconnectivity of the associated conditions under the umbrella term "metabolic syndrome", treating each at once remains a complex scientific enigma [16]. As such, experts recommend that physicians investigate these individually for now, given that no drug has yet addressed the syndrome holistically [17]. Although pharmacological approaches induce definitive and instantaneous effects in the prevention of each linked disease, therapeutic lifestyle changes (TLC) persist in being the core of clinical prophylaxis for metabolic syndrome, mainly due to the side effects imposed by drug intakes [18], making these less executable in the long-term. A cause célèbre is the introduction of alkaline ionized water (AIW), as it entails a propitious solution or, at the very least, provides a lens that might help in procuring a viable cure for metabolic syndrome.

Alkaline ionized water (AIW), also known as alkaline reduced water (ARW), is reported to benefit one's health. ARW can be produced via the electrolysis of water, and exhibits an alkaline pH between 8 and 10. ARW is known to have a protective effect on the deoxyribonucleic acid (DNA) against oxidative damage, since it can scour reactive oxygen species (ROS), reducing inflammation, and thereby resulting in disease prevention [19]. A study on the clinical effects of ARW and how it works demonstrated that the mechanism is due to its negative oxidation potential and abundant dissolved hydrogen [20]. Interestingly, a study on longevity revealed that ARW increased survival functions, resulting in a longer lifespan [21]. With the increasing population of people suffering from lifestyle diseases and metabolic syndrome that may reduce not only the quality of life, but also the individual's lifespan, the use of ARW may be a potential intervention to combat these diseases. With this, this review mainly contains literature acquired with the keywords "alkaline ionized" (or "reduced") in combination with the following words: "lifestyle disease"; "metabolic syndrome"; "obesity"; "cancer"; "diabetes"; "microbiota"; and "exercise". Information from journal articles and case studies was indexed and obtained from several databases, and presented data were produced from selection processes.

## 2. Metabolic Syndrome and ARW

### 2.1. Obesity

Obesity is characterized by the accumulation of adipose tissue that impairs physiologic processes. Moreover, it is a disorder concerning energy regulation, and is connected to several diseases which constitute metabolic syndrome [22]. Being a disorder of energy regulation, obesity affects both energy expenditure and food intake. Furthermore, the control of consumption and expenditure is governed by nervous and endocrine components, where neural parts respond appropriately to the signals supplied by hormones and other substances. These substances may include leptin, ghrelin, and peptide YY (PYY). In individuals and model animals, such as mice, obesity results from the loss of either leptin or its receptor, affecting the leptin pathway (Figure 2). Although rare, the loss of leptin or its receptor causes massive obesity in both mice and humans. From patients with extreme obesity, about 4% to 5% were observed to have a mutation in the melanocortin receptor 4 genes (*MC4R*). In addition, obesity is also accompanied by a low-grade chronic inflammation marked by increased immune markers (such as TNF, IL-1, IL-6, and IL-18), chemokines, and steroid hormones [22–24]. Immune cell imbalance within adipose tissue

may be blamed for the development of obesity together with systemic inflammation [23,25]. The etiology of obesity is not entirely understood, although family history, and environmental and psychological factors are involved. Moreover, obesity is commonly associated with type 2 diabetes mellitus, dyslipidemia, cardiovascular diseases (such as hypertension), and some cancers [22].

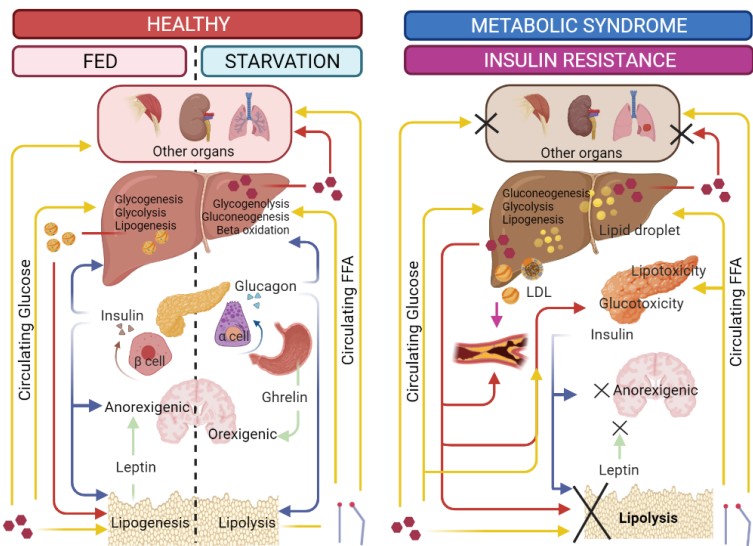

**Figure 2.** A comparison of metabolic processes in healthy people and those with metabolic syndrome. Insulin controls a variety of metabolic processes in the body. Its primary function is to keep systemic glucose levels under control. During the fed state, glucose is absorbed by the intestine, increasing circulating nutrients, such as glucose, amino acids, and lipids. The beta cells of the pancreas produce insulin, which aids in the uptake of glucose in various tissues and organs. Insulin promotes glycogenesis, glycolysis, and lipogenesis in the liver. This also boosts the production of high-density lipoprotein (HDL) in the liver, which facilitates lipid trafficking. Insulin stimulates lipogenesis in adipose tissue, which causes it to secrete leptin. Leptin and insulin stimulate anorexigenic signaling in the brain, which helps to regulate appetite. On the other hand, fasting and starvation reduce insulin levels, resulting in increased lipolytic activity in adipose tissues. The secretion of glucagon by alpha cells promotes lipolysis in adipose tissue, increasing the amount of free fatty acids in circulation for the liver and other organs. Insulin resistance, on the other hand, inhibits glucose uptake by tissues and organs, causing blood glucose levels to rise. It cannot antagonize lipolysis in the adipose tissue, resulting in more free fatty acids entering the bloodstream. High levels of glucose and fatty acids induce oxidative stress and toxicity to tissues and organs. Fatty acids and glucose enter the liver, promoting lipid droplet generation, which eventually leads to stenosis. Further, LDL production in the liver is upregulated, increasing the risk of cardiovascular diseases. Insulin resistance also activates the orexigenic center of the brain, which causes a person to have an increased appetite. (Created with BioRender.com).

Treatments for obesity range from medications and lifestyle changes, to surgeries. However, conventional drugs to treat the disorder are limited due to their side effects [26]. Recently, several studies have shown the effectiveness of ARW consumption in clinical and animal models (Table 1). In particular, the study of Kamimura et al. examined the effects of long-term *ad libitum* drinking of ARW on OB-R deficient mice (db/db), which demonstrated an increase in energy metabolism and control of fat and body weight [27]. The study also elucidated upregulation of the hepatic hormone, fibroblast growth factor 21 (FGF21), which promotes glucose and fatty acid expenditure. Moreover, an increase in cytochrome P450 (CYP7A1) expression was observed in C57BL/6 mice with induced obesity through a high-fat diet, along with regulation of body weight increase and amelioration of adiposity, epididymal, and liver fats upon the administration of ARW [28]. The upregulation of CYP7A1 is believed to be a response to an increasing cholesterol catabolic rate. In addition,

consumption of ARW by Sprague-Dawley rats decreased bodyweight, serum triglycerides, and fatty liver deposition [29].

## 2.2. Diabetes mellitus

Diabetes mellitus is a collection of metabolic illnesses sharing the common feature of hyperglycemia caused by abnormalities in insulin secretion, insulin action, or, most commonly, both [30]. Chronic hyperglycemia and related metabolic dysregulation may be associated with secondary damage in various organ systems, including the kidneys, eyes, nerves, and blood vessels. Diabetes and associated disorders of glucose metabolism are widespread. Around 422 million people have diabetes worldwide, with India and China being the major contributors to the world's diabetic burden, according to WHO [1,2]. Increasingly sedentary lives and poor eating habits have contributed to the simultaneous ascent of Type 2 diabetes and obesity, which some have termed "the diabesity epidemic". Sadly, this sickness has already stretched to children living in "food deserts" who depend on highly processed diets heavy in carbohydrates and sugar, and do not exercise adequately [22].

Although hyperglycemia is a common symptom of all kinds of diabetes, the underlying conditions that cause it differ widely. Previous diabetes categorization methods were based on clinical characteristics, such as the age of onset of disease and the form of therapy. However, the new classification includes a better understanding of the pathophysiology of each variety. The great majority of diabetes cases fall into one of two categories: Type 1 diabetes (T1DM) or type 2 diabetes (T2DM). Type 1 diabetes, or T1DM, is an autoimmune disease in which immune effector cells assault endogenous β-cell antigens, causing islet loss. T1DM is most typically diagnosed in adolescence, arises during puberty, and worsens with age. The term "juvenile-onset diabetes" is no longer used because the disease can strike at any age, even in late adulthood. Similarly, because many forms of diabetes eventually require insulin treatment, the term "insulin-dependent diabetes mellitus" has been deleted from the current categorization of diabetes. Despite this, most T1DM patients need insulin to live, and without it, they risk experiencing significant metabolic consequences, such as ketoacidosis and coma. T1DM, like most autoimmune illnesses, is caused by a mix of inherited and environmental factors.

Meanwhile, type 2 diabetes, or T2DM, is a complex disease that involves the interplay of hereditary and environmental variables, and a pro-inflammatory state. Unlike T1DM, there is no evidence of an autoimmune basis. Genetic predisposition contributes to the pathophysiology, as indicated by the illness concordance rate of greater than 90 percent in monozygotic twins, a percentage higher than in T1DM. Furthermore, first-degree relatives have a 5- to 10-fold higher risk of having T2DM than individuals without a family history, when matched for age and weight. Many of these genes are implicated in adipose tissue function (via impacts on bodily fat distribution (visceral vs. subcutaneous)), islet β-cell activity, and obesity. It is considered that when combined, these genetic polymorphisms conspire to give the genetic basis for T2DM risk. However, the heritable risk remains a small component of determining illness vulnerability, and environmental factors are the key contributors. Insulin resistance predates the development of hyperglycemia, and is frequently accompanied by compensatory β-cell hyperfunction and hyperinsulinemia in the early stages of the genesis of T2DM. Over time, the inability of β-cells to adjust to the increased secretory demands for maintaining a euglycemic state results in chronic hyperglycemia and the long-standing consequences of diabetes. While insulin resistance by itself can lead to decreased glucose tolerance, β-cell dysfunction is a prerequisite for the development of overt diabetes (Figure 2). In contrast to the severe genetic abnormalities in β-cell function that occur in monogenic forms of diabetes, the β-cell function increases early in the disease process in most people with "sporadic" T2DM as a compensatory effort to reverse insulin resistance and preserve euglycemia. Eventually, however, β-cells seemingly exhaust their capacity to adjust to the long-term demands posed by insulin resistance, and the hyperinsulinemic condition gives way to a state of relative insulin deficiency, that is, insulin levels are insufficient for blood glucose. Several processes

(Figure 2) have been implicated in causing β-cell dysfunction in T2DM, including the following: lipotoxicity; glucotoxicity; incretin effect; amyloid deposition inside islets; and genes that affect insulin production [27].

In relation to the possible effects of ARW on diabetes, it can be further expounded by different studies worldwide (Table 1). First, a study by Kamimura et al. experimented on obesity model mice (db/db) with lacking functional leptin receptors [27]. Using commercially available assays, the amounts of ketone bodies, triglycerides (Tri), and total cholesterol, HDL, LDL, glucose, and non-esterified FA in the blood were determined, and by using an open-circuit calorimeter, indirect calorimetric studies were conducted to determine metabolic rate. Drinking $H_2$ water was found out to stimulate energy metabolism. This resulted in alleviated levels of plasma glucose, triglycerides, and insulin, due to the upregulation of FGF21, a hepatic hormone that increases glucose and fatty acid expenditures [31]. Furthermore, a study on Otsuka Long-Evans Tokushima Fatty (OLETF) rats given ARW showed that they had considerably lower blood sugar levels than those given tap water alone. Moreover, triglycerides and total cholesterol were shown to be lower in the ARW group than in the control group, demonstrating that ARW acts as an antioxidant against ROS produced by the non-phagocytic NAD(P)H oxidase protein within blood vessels [32]. On the other hand, based on the clinical study of Siswantoro et al., in individuals with type 2 diabetes, drinking water with a pH of 9.5 or 11.5 has been found to help reduce blood sugar levels. Because free radicals are produced by hyperglycemia, water with an alkaline pH acts as an antioxidant. These free radicals receive electrons from alkaline water [33]. Moreover, consumption of ARW had a synergistic impact on reducing oxidative markers and increasing quality of life when combined with walking. Hydrogen diffuses readily into the cell, where it binds to hydroxyl radicals. Inflammation, which is a consequence of oxidative stress, was also found to lessen when AEW was taken—it lowered IL-6, TNF, and IL-12, as well as IFN-gamma [34]. Finally, it resulted in increased plasma levels of adiponectin and extracellular-SOD in a clinical experiment with 30 T2DM patients who had undergone diet change and exercise therapy, and six persons with impaired glucose tolerance for eight weeks [35].

**Table 1.** Summary of ARW studies in metabolic syndrome.

| Metabolic Syndrome | Model Used | Results | Possible Mechanism | Reference |
|---|---|---|---|---|
| Diabetes | db/db Obesity Model Mice | 1. Hepatic oxidative stress was reduced. 2. Significantly decreased fatty liver in a.) db/db mice and in b.) wild-type mice. 3. Long-term consumption led to controlled fat and body weight despite no change in diet. 4. Alleviated levels of plasma glucose, insulin, and triglycerides. 5. Drinking $H_2$ water stimulates energy metabolism. | Upregulation of hepatic hormone, FGF21, which increases glucose and fatty acid expenditures. | [27] |
| | OLETF Rats | 1. The bodyweight of the ARW group has significantly increased compared to the control. 2. Rats treated with ARW had a significantly decreased blood sugar level than those treated with tap water only. Moreover, triglycerides and total cholesterol were lower than the control group. 3. GPT and GOT in both groups had significant differences. | ARW functions as an antioxidant against ROS generated within blood vessels by the non-phagocytic NAD(P)H oxidase protein. | [32] |
| | Clinical | 1. Consumption of water with pH 9.5 or 11.5 is observed effectively in reducing blood sugar levels of those who have type 2 diabetes. | Water with alkaline pH acts as an antioxidant against free radicals generated by hyperglycemia. Alkaline water donates electrons to these free radicals. Free radicals react with cellular unsaturated fatty acids that cause the production of harmful peroxides that damage the cells. | [33] |

**Table 1.** *Cont.*

| Metabolic Syndrome | Model Used | Results | Possible Mechanism | Reference |
|---|---|---|---|---|
| | Clinical | 1. Consumption of ARW (together with walking) synergistically lowers oxidative markers and improves the quality of life. | ROS scavenger (antioxidant), which reduces inflammation and protects DNA from damages. Hydrogen readily diffuses into the cell, which binds to hydroxyl radicals. Inflammation, being a product of oxidative stress, was also seen to decrease upon treatment of ARW—reduces IL-6, TNF, and IL-12, as well as IFN-gamma. | [34] |
| | Clinical | 1. Hydrogen-rich water significantly decreases levels of modified LDL, small dense LDL, urinary-8-isoprostanes, serum concentrations of oxidized LDL, and FFA. 2. It also resulted in increased plasma levels of adiponectin and extracellular-SOD. | Hydrogen suppresses oxidation/modification of LDLs. | [35] |
| Obesity | Sprague-Dawley rats | 1. Bodyweight of MRW-treated rats was lower. 2. Serum level triglycerides and liver fat deposition in rats treated with MRW were significantly reduced. | MRW downregulates lipid-metabolism in the liver, thus suppressing obesity—lowers serum TG. Hydrogen acts as an antioxidant against ROS. | [29] |
| | C57BL/6 Mice | 1. Compared to the control group, consumption of ARW decreased adiposity, controlled body weight gain, reduced accumulation of epididymal fats, and decreased liver fats. 2. Moreover, levels of adiponectin and leptin were well coordinated. 3. Upregulation of cytochrome p450 was observed. | CYP7A1 induces higher cholesterol catabolic rate. | [28] |
| Non-alcoholic fatty liver disease | C57BL/6 High-Fat Diet Model | 1. Decreased fat mass of HRW-treated mice. 2. Less hepatic lipids and mild steatosis were observed in HRW-treated mice compared to control. | CD36 and TNF -$\alpha$ expression levels in the liver are significantly reduced in HRW-treated mice compared to control. | [36] |
| | High-Fat Diet and Streptozotocin-induced Lipotoxicity and Glucotoxicity in Sprague-Dawley Rats | 1. Less steatosis and apoptotic hepatocytes in the HRS group. 2. The insulin resistance index is low in the HRS group compared to the disease model group. | PPAR$\alpha$ and PPAR$\gamma$ protein expression in the liver were upregulated in the HRS group. TNF-$\alpha$ and IL-1$\beta$ were significantly decreasing in the HRS group. | [37] |
| | Methionine-choline–deficient (MCD) diet C57BL/6 Mouse Model | 1. Reduction in hepatocyte apoptosis in HRW treatment. 2. Decreased in hepato-cholesterol and increased antioxidant. | Inflammatory markers TNF-$\alpha$ and IL-6 were downregulated. Lipid metabolism markers FAT and AOX were significantly reduced in the HRW group. | [38] |
| | Stelic Animal Model (STAM) of NASH-derived HCC | 1. Less steatosis and inflammation in the liver of HRW-treated mice. 2. Tumor burden is significantly lower in the HRW group. | | |
| Cancer (Skin) | C56BL/6 Mice B16-BL6 Melanoma Cells of Mice | 1. Reduced number of B16-BL6 melanoma cell colonies. 2. Longer survival span and delay in the tumor growth. 3. Positive immune-modulating effect: systemic cytokines and cytokines for both cellular immunity and humoral immunity were stimulated. 4. Reduced amount of ROS in lung, liver, and kidney except for the spleen. | Inhibition of intravenous metastasis. ARW acts as an antioxidant by reducing the amount of ROS, and functions as a strong immune modulator. Magnesium ion components of ARW might aid in its anticancer effect. | [39] |
| Cancer (Lung) | A549 Cell Line | 1. Intracellular $H_2O_2$ levels were reduced. 2. ERW inhibits VEGF gene expression and extracellular secretion in tumor cells. ERW regulates VEGF gene transcription. 3. ERW treatment of A549 cells resulted in a decreased total tube length and reduced in all parameters. | Activation of hydrogen with enhanced reducing potential produced during electrolysis of ERW aid in the scavenging of reactive oxygen species. ERW blocks ERK activation in A549 cells. Inactivation of ERK that leads to ERK MAPK signal pathways by ERW is suggested to play a pivotal role in inhibiting VEGF gene expression. | [40] |

**Table 1.** *Cont.*

| Metabolic Syndrome | Model Used | Results | Possible Mechanism | Reference |
|---|---|---|---|---|
| Cancer (Human Fibrosarcoma) | Human Fibrosarcoma HT1080 Cells | 1. ERW was effective in decreasing the concentration of intracellular $H_2O_2$ in HT1080 cells. 2. Significant decrease in invasive activities of the HT1080 cells treated with ERW and AERW. 3. ERW reduced gene expression MMP-2 and MT1-MMP gene more than the AERW. ERW attenuated gene expression of MMP-2 induced by excessive $H_2O_2$. ERW also inhibited MMP-2 activation induced by H2O2 and PMS. 4. ERW inhibits MMP-2 gene expression via P38 MAPK inactivation. | The ROS scavenging activity of ERW can be attributed to its two active substances: hydrogen molecules that protect from free radicals by enhancing the expression of genes encoding antioxidant proteins (SOD, catalase, and HO-1 enzymes); and platinum (Pt) nanoparticles that can scavenge $O_2^-$, $H_2O_2$, and OH radicals. ERW inhibits invasion of the HT1080 cells through a high reducing potential of its hydrogen molecule component, and the ROS scavenging ability of Pt nanoparticle component. ERW has an antagonizing effect on the amplified activation of p38 due to $H_2O_2$ treatment. | [41] |

### 2.3. Non-Alcoholic Fatty Liver Disease

Non-alcoholic fatty liver disease (NAFLD) encompasses a wide range of liver conditions, including non-alcoholic fatty liver (NAFL), non-alcoholic steatohepatitis (NASH), advanced fibrosis, and end-stage liver disease, as well as hepatocellular cancer [42]. It is the most common liver-related metabolic syndrome, afflicting one-third of the world's population [43]. Inflammation, nutrient and energy homeostasis, genetic background, microbiota, and lifestyle are some factors that can draw on the pathological triggers of NAFLD [44]. Fat buildup in the liver of patients who have NAFLD can be caused by abnormal levels of free fatty acids (FFA) in the blood [43]. Insulin controls FFA levels, and consequently, plays a role in the onset of metabolic syndrome [45]. Insulin has an antilipolytic effect that persists after feeding [43]. However, the breakdown of lipids in adipose tissue increases during fasting to provide nutrients to organs, except the brain. When adipose tissue develops insulin resistance, it aberrantly secretes FA, increasing the amount of circulating FFA [46], as illustrated in Figure 2. Excess FFA is absorbed by various tissues and organs, and the positive energy balance induces lipid droplet accumulation in the cells, resulting in lipotoxicity and cellular dysfunction [47]. Excessive lipid droplet accumulation in NAFL patients can result in oxidative stress, inflammation, and hepatocyte injury, eventually leading to NASH [40]. Because NAFLD is a multifactorial disease, different mouse models are used to study NAFLD pathogenesis, including methionine-choline diet-induced NASH, high-fat diet-induced NAFLD, and streptozotocin-induced NASH-related hepatocarcinogenic models [27,36–38,48]. Several of these studies used hydrogen therapy, which demonstrated a therapeutic effect against NAFLD (Table 1). $H_2$ intervention significantly reduced oxidative stress, inflammation, and steatohepatitis. Supplementation of hydrogen-rich water (HRW) in HDF mice abolished steatosis by decreasing redox imbalance in hepatocytes and increasing hepatic FGF21, which positively regulates energy homeostasis [27]. Interestingly, HRW treatment also modulates transcriptional changes in lipid metabolic genes by increasing peroxisomal fatty acyl-CoA oxidase (ACOX), a rate-limiting enzyme for beta-oxidation of very-long-chain fatty acids. It also inhibits the transcription of CD36, a key protein in fatty acid uptake [38].

### 2.4. Cancer

Cancer is defined as a multitude of diseases wherein cells abnormally proliferate and can spread to infiltrate nearby normal body tissues [49]. It involves many disorders with varying origins, progression, treatment response, natural histories, intensity, and magnitude. The physiological processes involved in cancer development are complicated, but genetic mutations (e.g., deletions, amplifications, and translocations) have been considered to play pivotal roles in cancer etiology [50]. These mutations result in the formation of

oncogenes and a non-functional set of tumor suppressor genes that allow cells to divide indefinitely [51]. The cause of these genetic changes can either be intrinsic, from random errors in DNA replication, inherited mutations, or extrinsic processes through acquired mutations from exposure to harmful substances in the environment (e.g., carcinogens, mutagens, UV, and ionizing radiation) due to lifestyle behaviors [52]. Examples of lifestyle behaviors that are cancer risk factors include: cigarette smoking; alcohol consumption; diet; infectious agents; and even genetic factors, such as tylosis [53–57].

Cancer cells are able to manufacture their own growth signals, overexpress growth signal receptors, and alter intracellular pathways involved in transducing signals for rapid cell growth [58]. In response to internal stress due to either oxidative damage from reactive oxygen species, or uncontrolled proliferation of cells with damaged DNA material, cells usually undergo apoptosis or cell death. Unfortunately, cancer cells evade apoptosis by inactivating a vital apoptotic protein called p53 protein [59]. These tumor tissues are highly associated with the progression of diseases, such as cachexia, by releasing factors that promote weight and muscle loss [60]. Since tumors cannot grow beyond the 100 µm size of capillaries, the formation of new blood vessels for oxygen and nutrient supply is sustained in cancer through upregulating angiogenesis-initiating signals (e.g., VEGF and FGF) [61]. Cancerous tumors can then metastasize and colonize new environments in the body, therefore, spreading cancer to different body parts while impairing their normal functions [62]. This process accounts for the death of the majority of cancer patients [63]. The ability of cancer cells to invade adjacent tissues is a function of epigenetic changes of proteins like c-Met, EGF receptors, RAS proteins, and PTEN [64].

Cancer treatment depends on the cancer type, stage, medical history, and status of each patient, but the most common treatment patterns involve surgery and a series of systemic treatments, such as chemotherapy, immunotherapy, and targeted and hormonal therapy, with biopsy being one of the definitive diagnostic procedures [65]. However, studies have revealed the potential role of alkaline reduced water in treating various cancers (Table 1). In both mouse and human cell line models, ARW was found to be effective in reducing the factors that are involved in cancer progression. For example, gene expression of VEGF and MMP-2 is downregulated in human cell lines induced in electrolyzed reduced water (ERW) and tested in vitro from the lung adenocarcinoma (A549) and fibrosarcoma (HT1080) cell lines, respectively [33,34]. The antimetastatic and tumor suppression capability of ARW was effective in B16-BL6 melanoma cells of mice tested in vivo, wherein cancer cell colonies were reduced, and a delay in tumor growth was observed [39]. Furthermore, experiments in both cancer cell lines of human and mouse models revealed the efficiency of alkaline induced water in decreasing the concentrations of reactive oxygen species, such as hydrogen peroxide, in the lung, liver, and kidney [39], as well as intracellularly [40,41].

From the previously stated risk, the host tissue microbiota plays a significant role in cancer development as well. The gut microbiota refers to the symbiotic microorganisms residing within the gastrointestinal tract [66]. These microorganisms defend the gastrointestinal tract against several pathogens. However, upon disruption (i.e., microbial dysbiosis), these may bring diseases and disorders [66]. Studies have found that changes in the gut microbiota may lead to liver cancer brought by unregulated immune response and the production of toxic bacterial markers, such as lipopolysaccharides and lipoteichoic acids [67,68]. Colon cancer is one of the cancers arising within the gastrointestinal tract as well. Aside from the lifestyle and genetic factors, colon cancer development is affected by the patient's diet. In addition, diet changes modify microbial population, and several species of the gut microbiota have been linked to the development of colon cancer, including infections brought by *Escherichia coli, Bacteroides fragilis,* and some species of *Streptococci* [69–71].

These microorganisms affect cancer development in several mechanisms, but mainly due to chronic infections [66,72]. Chronic infections promote cell transformations, genetic alterations, and hormone deregulation [62,66,69,73]. Bacterial dysbiosis strengthens the

effect of chronic infections by promoting DNA damage by increasing concentrations of immune markers IL-1,6, and 10, together with an increase of TNF-$\alpha$. Increase of these immune markers modulates a stepwise process of stimulating NF-$\kappa$B, Wnt signaling, and MAPK signaling. Overall, these steps inhibit apoptosis and increase oxidative substance formation [72,74–76].

The human body is a microbial ecosystem that harbors a diversity of microorganisms, including bacteria, viruses (prokaryotic and eukaryotic), fungi, and protozoa [77,78]. Although the presence of microbes within the human body usually connotes the presence of disease-causing agents or pathogens, studies have shown that the relationship between mammals and their microbiomes have shifted from being commensal to symbiotic, such that their absence or an alteration in their composition are associated with less efficacious nutrition, immunological response, metabolism, and resistance to pathogen colonization [79–81]. Furthermore, the regulation of cognitive activity and behavior through mechanisms of communication is achieved by the concept of a microbiome-gut-brain axis, which constitutes the exchange of information between the central nervous system and the gut via neurotransmitters and metabolites (short-chain fatty acids—SCFAs) generated by the gut microbiota itself [82,83]. As such, researchers have been trying to utilize this aspect by studying the implications of drinking alkaline reduced water (ARW) to the gut microbiome, and whether the outcomes of such studies can be beneficial to human health, more specifically, in treating metabolic diseases.

Abnormal changes in the composition of the gut microbiota in both human and animal models have been linked to obesity, insulin resistance or type 2 diabetes mellitus (T2DM), chronic kidney disease, hypertension, and many other cardiometabolic sequelae [39,84–86]. On obesity, it was reported that great amounts of SCFAs were found in the feces of obese patients [82], however, it was contradicted by another study that shows a great amount of SCFAs from the fecal samples of lean individuals [87]. Using mice as a model, another study, which hypothesized about the antioxidative properties of hydrogen-rich/hydrogen-dissolved ARW, concluded that oral ingestion of ARW alters the relative abundance of some of the gut microbial taxa and accordingly changes cecal metabolite contents, particularly with an increase in the production of propionic acid, isobutyric acid, and isovaleric acid, whereas no changes were found for the remaining intestinal acids, which include butyric acid, succinic acid, formic acid, lactic acid, acetic acid, or valeric acid [88]. This indicates a greater potential in colonic fermentation. Of the acids that had shown an increase in production, researchers are eyeing propionic acid, which improves insulin sensitivity and inhibits lipolysis, from which humans could take advantage of in treatments for T2DM and obesity, respectively [89–91].

As opposed to these findings, a study in 2018 reported that drinking ARW found no significant change in the gut microbiota and glucose regulation, sampled from 24 volunteers who completed the interventions [92]. In their experiment, discrepancies may have rooted from its design wherein healthy, non-drinking volunteers were allowed to maintain usual diet routines, which could have possibly shrouded the potential effects of drinking alkaline water. This was further backed by another study using murine models, which discussed how drinking $H_2$-rich water demonstrated more positive effects than that of drinking ARW in mice with non-alcoholic fatty liver disease (NAFLD), manifesting less weight gain, even in a high-fat diet [36]. Interestingly, results of other research suggest that habitual drinking of ARW relieves symptoms of gastrointestinal disorders, renormalizes stool consistency [93–95], and acts upon a specific bacterial taxon *Bifidobacterium* spp., which exhibits an increase in relative abundance [96]. This genus is highly anaerobic, and thus, further supports the notion of $H_2$-rich ARW having antioxidative properties.

With the help of modern technology, the role of the gut microbiota in metabolic disorders has now become a clearer picture [97]. The role of drinking ARW in the gut microbiota, on the other hand, has shown a lot of promise, however, it is still under further probing.

### 3. Nutritional and Therapeutic Interventions for Metabolic Syndrome

*3.1. Effects of ARW on Diabetes, Obesity, and Exercise*

The discovery of ARW dates back to 1990, with most articles coming from Japan. ARW was found to be advantageous in various fields, including agriculture, biochemistry, molecular biology, and engineering. From Scopus, 135 journal papers dealing with the topic have been published up to this point. In the titles or abstracts of these papers, phrases like "Alkaline Ionized Water", "Alkaline Reduced Water", "Alkali-Ionic Water", "Alkaline Induced Water", "Alkaline Hydrogen Water", and "Alkaline Electrolyzed Water" have appeared (Figure 3). The inclusion and exclusion parameters that were used to select the Scopus-articles cited in Figure 3 and summarized in Table 1 revolved around the study design, date, and type of publication. Initially, articles screened using Scopus that focus on ARW, or similar terms with relation to diseases associated with metabolic syndrome, were filtered by excluding review articles and studies with samples having pathogenic and genetically acquired diseases. Instead, articles that employed human clinical trials and in vitro experiments on study models involving animals, humans, and cell lines were chosen. To further filter, articles with a publication date before 2005 were removed from the selected list. This review highlights the impact of ARW on metabolic illnesses. Here, the intervention of ARW towards diabetes has been recorded in a total of seven articles in the last couple of years, notably 2006, 2012–2014, 2016, and 2020, the most numerous among other metabolic syndromes. There were four cancer-related ARW publications and three obesity-related ARW articles. Finally, only one ARW study in relation to NAFLD was accounted since 2018.

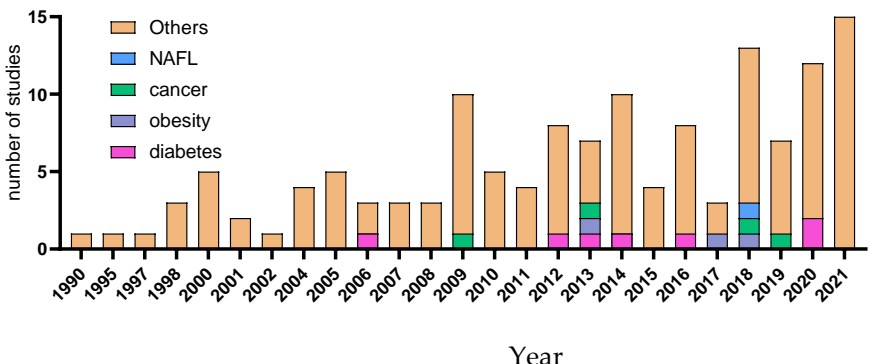

**Figure 3.** Frequency of published studies on alkaline reduced water by subcategories of metabolic syndrome. Journal articles that contain terms related to ARW and metabolic syndrome were indexed from Scopus. Abbreviation: non-alcoholic fatty liver, NAFL.

As previously mentioned, diabetes mellitus (DM) is a collective term for a metabolic syndrome characterized by hyperglycemia due to aberrations in insulin secretion, insulin action, and/or insulin tolerance [1,2]. This was said to be related to the levels of ROS in the body, specifically to the blood vessels, tissues, and cells. Controlling lipid metabolism was also significant in studying diabetes [32]. More so, other related diseases can also appear in relation to the effects of several metabolic, hormonal, oxidative stress, and/or immunological imbalances. However, due to the costly maintenance medicine for patients that are diagnosed with this metabolic syndrome, alkaline reduced water (ARW) exhibiting a pH between 8 and 10 was used as an alternative [32,33], and hence, several studies investigated the effects of ARW towards several metabolic syndromes and related diseases.

One such study is an experimental study conducted on OLETF rats. It was observed that the rats treated with ARW had significantly lower blood sugar, triglycerides, and total cholesterol levels than the control group treated with tap water only. The bodyweight of the ARW group was also significantly higher than the control group. The results were attributed to the antioxidant properties of ARW, involved in lipid metabolism and body weight changes, that exhibit an increased superoxide dismutase activity due to the in-

creased dissociation activity of ARW. Furthermore, the scavenging activity of ARW was attributed to the abundance of activated dissolved hydrogen. When cardiac muscles partially degenerate due to the lack of oxygen caused by lipid blockage, glutamic oxaloacetic transaminase (GOT) and glutamic pyruvic transaminase (GPT) are secreted into the blood. However, GOT and GPT levels were observed to be significantly lower in the ARW group than those of the control group, suggesting that ARW can prevent heart diseases that may further cause diabetes [32]. Additionally, the upregulation of the hepatic hormone, FGF21, significantly increases glucose and fatty expenditures, thereby alleviating plasma glucose, insulin, and triglycerides [27]. These results also coincided with another quasi-experimental study on diabetes mellitus patients. After ARW administration, blood sugar levels lowered among patients with T2DM. This was also attributed to the antioxidant properties of ARW, which can donate electrons to free radicals generated by hyperglycemia. The free radicals that are formed, which weaken antioxidant systems, react with the unsaturated fatty acids in the cells that cause the production of harmful peroxides that damage the cells. Additionally, due to the relatively smaller water molecules of ARW containing bicarbonate ions, it can diffuse easily in the body cells. Specifically, it can be utilized easier by the pancreas, thereby increasing its ability to maintain the alkalinity in the body, and improving its insulin-production process [33,98].

Along with the increased risk of diabetes, there is also the increased risk of obesity or dyslipidemia, as the two metabolic syndromes were commonly associated with each other [22]. Hence, alkaline ionized water was also observed to decrease total cholesterol levels and improve other lipid profiles in T2DM patients that also have dyslipidemia [98]. Moreover, it was also suggested that ARW is effective against obesity or the accumulation of adipose tissue that may impair physiological processes in the body [22]. Oxidative stress was, again, further suggested to intensify obesity, increasing the accumulation of fat. When an adipose tissue experiences oxidative stress, the expression of the peroxisome proliferators-activated receptor (PPAR)-λ and adiponectin, an adipocytokine that suppresses insulin resistance and dyslipidemia, downregulate. However, upon administration of ARW, water generated by reacting with an alkaline earth metal, obesity was neutralized and suppressed. Because ARW downregulates lipid-metabolism in the liver, it can lower the TG levels in the blood. ARW also increases the expression of adiponectin. This was observed in an in vivo study conducted using high-fat fed SD rats, which showed significant results in the anti-obesity effects of ARW [29]. The results also correspond to another study involving a high-fat diet in mice. Consumption of ARW led to a decrease in the adiposity of the body and liver fats, and controlled body weight gain. As previously discussed in earlier sections, the loss of either leptin or its receptor results in adipose tissue dysregulation. However, in the study of Jin et al., 2006 [29], levels of adiponectin and leptin were coordinated. Further, CYP7A1, which was a key regulator in bile acid biosynthesis encoding the rate-limiting enzyme, was observed to be significantly higher in mice that consumed ARW, suggesting that alkaline reduced water induces higher cholesterol catabolic rate [28].

To further ameliorate the oxidative stress causing diabetes and obesity, employing regular (aerobic) exercise was further suggested. Exercise refers to any physical activity to improve or maintain physical fitness. In employing such physical activity, the muscles and organs of the body produce heat, which must be eliminated via sweating to prevent hypothermia [99]. Hence, a proper hydration state should be observed to replace the fluids lost (~2% to 3% of the body mass) [100]. Since water is the most widely used fluid, its different properties, mineral content, acidity, and/or basicity (pH) can determine hydration status and other therapeutic properties during exercise. During an exercise, especially one that is considered supramaximal, there are subtle changes in blood and tissue pH that increase the proportions of ROS and RNS [101]. Various fluid replacements may be used depending on the type of exercise being performed. In terms of salt and sugar concentrations relative to the human body, most drinks are isotonic (equal concentrations), while some have higher (hypertonic), and/or lower concentrations of salts and sugar (hypotonic) [102].

Several studies also found beneficial effects of ARW on exercise, enabling the maintenance of muscle performance, as well as the reduction-oxidation self-regulating process to maintain internal stability (homeostasis) during consecutive days of exercise [34]. The biochemical reactions that occur inside the human body are mostly pH-sensitive, hence, drinking alkaline water may facilitate most of these reactions and accelerate the body's recovery process after an exercise [103]. Although ARW and exercise were suggested to ameliorate oxidative stress and inflammation, there have been limited studies conducted to investigate their combined effects on alleviating inflammation, oxidative stress, hydration status, and, using urine analyses, acid-base equilibrium responses.

In a randomized control study, synergistic effects of drinking ARW and walking as an exercise were investigated among patients with T2DM. The results showed that the simultaneous utilization of exercise and ARW independently reduced oxidative stress biomarkers, such as fasting blood glucose and malondialdehyde, advanced glycation end products, and advanced oxidation protein products. It also reduced inflammatory biomarkers that can be detected in T2DM patients, such as the white blood cell count and the ratio between neutrophils and lymphocytes. The scavenging ability of ARW was credited to its active atomic hydrogen, produced during water electrolysis, which possesses a high reducing ability which could engage in reactions involving reduction-oxidation, leading to an increase in antioxidant levels [34]. Its consumption also prevents the production of ROS when performing high-intensity exercise [104,105]. Metabolic acidosis that occurs during high-intensity exercise, and aerobic and anaerobic exercise, involves the production of lactic acid, reducing blood pH, and altering acid-base equilibrium. Studies demonstrated that drinking alkaline water improves acid-base balance [101,104]. Furthermore, during a high-intensity exercise, rapid adenosine triphosphate (ATP) hydrolysis occurs, resulting in the formation of diphosphate and adenosine monophosphate (AMP). The amino groups contained in these structures were removed (in a process called deamination) and converted into inosinomono-phosphate in the purine nucleotide cycle [101]. After an intensive exercise, drinking ARW also helps restore the equilibrium via the exchange of ions, carbon dioxide, and water molecules occurring between inside and outside cellular components. These effects are due to the increase in glycolytic ATP production, which was attributed to the greater muscle buffering capacity and enhanced removal of protons [104]. The disturbances occurring in the fluids containing electrolytes were also reduced, along with the observed speed-up in the utilization of lactic acid [101]. Furthermore, therapeutic and control studies investigating the hydration status of athletes indicate that alkaline low mineralized water, and its habitual consumption, lead to a decrease in the specific gravity of the urine, a rather favorable reaction towards an intensive/supramaximal exercise [101,103,104]. Alkaline water with a pH between 8 and 10 has positive feedback on diabetes and obesity, as well as a synergistic effect on exercise. It can reduce oxidative stress and inflammatory biomarkers, and scavenge and inhibit the production of ROS, thereby increasing antioxidant levels. It also improves hydration status and acid-base equilibrium, reducing the disturbances in the electrolyte content of the fluids, speeding up the rate of lactate utilization. Alkaline water also improves aerobic and anaerobic performance.

### 3.2. ARW Mechanism of Action

Significant health threats involve the accumulation of ROS, accompanied by dysregulated lipid metabolism and inflammation, which are the primary risk factors for the increasingly prevalent lifestyle-related metabolic diseases [106]. From this perspective, it is important to explore the relationship between ROS scavenging, inflammatory control, and lipid metabolism. Research on alkaline ionized water has been increasing up to this date since its first development in Japan in the 1990s [107]. Nowadays, ARW has been applied in several fields, including the food industry, agriculture, and medicine. It has received tremendous attention because of its efficacious and novel medical treatment for various immune and oxidative stress diseases in China, Japan, and Korea [20,108,109]. ARW exhibits special properties, such as an alkaline pH, small-clustered water molecules,

negative ORP value, and a high content of dissolved hydrogen. Generally, ARW is formed by electrolysis, whereby water ($H_2O$) decomposes into oxygen ($O_2$) and hydrogen gas ($H_2$) due to an electric current. ARW generation has progressed and advanced in development. Most recently, it can now be generated using renewable energy [110]. ARW can also be generated with various additives, such as alkaline minerals and nanoparticles [111,112], to increase its antioxidant and therapeutic effects.

A number of studies have shown that ARW has antioxidant properties, and thus helps to counteract free radicals in the body. Free radicals can cause premature aging and other diseases and conditions. The chemical properties of ARW, especially its pH, are believed to contribute to its beneficial effects. ARW exhibits a higher pH than tap water, which plays an important role in its known efficacy. A previous study showed that ARW could efficiently induce $H_2O_2$ and $HOCl^-$ dependent antioxidant defenses, and reduce $H_2O_2$ and HOCl-induced oxidative stress [113]. With the mechanism mentioned above, supplementation with ARW has shown the following potential benefits: improvement of the health of the digestive tract [114]; alleviation of the severity of diseases in vivo [32,115]; and improvement the body condition of aged subjects [116]. However, the precise process and mechanisms whereby ARW contributes to disease improvement and prevention have not been fully understood. In line with this, the effects of ARW were confirmed, and its mechanism of action was assessed, in several models, including an animal and clinical model. Specifically, the potential benefits of ARW on the immune response were investigated, including its effect on the balance between Th1 and Th2 activation [116]. Some studies have suggested that ARW could cause changes in metabolic activity in vivo [117]. At the same time, ARW exhibits a higher pH than tap water, which plays an important role in its known efficacy. Drinking ARW would be beneficial not only in gastrointestinal diseases, but also in blood pH. It is known that the high pH of alkaline water, such as ARW, affects the clustering of water molecules, producing smaller water clusters that could enable the alkaline reduced water to efficiently enter the cell, increase intracellular hydration, flush out waste, and prevent unnecessary material from accumulating in the cell [118]. So far, these results have further strengthened the findings of Kim and Yokoyama [119], and Watanabe et al. [120], who found that long-term supplementation with ARW normalized abnormal blood glucose and lipid levels. In addition, Li et al. [121] recently reported that ARW prevented apoptosis of pancreatic β-cells, and that long-term ingestion of ARW slowed the development of symptoms in a mouse model of type 1 diabetes mellitus by preventing the alloxan-derived generation of reactive oxygen species. Previous studies have also demonstrated that ARW protects against the accumulation of lipids and cholesterol in the body [29,32]. Ignacio et al. [28,122] designed a study to further confirm the effects of ARW in diseases related to fat accumulation, such as obesity. It is believed that immune profiling, including Th1/Th2 network and cytokine profiling, is important in the treatment of inflammatory diseases, such as obesity [118]. Therapeutic interventions, such as ARW with specific anti-inflammatory/Th-2 type cytokines, hold promising proof as a treatment against the excess pro-inflammatory response in obesity. Alongside this, alkaline water facilitates lipid disintegration—indeed, supplementation with ARW protected mice from quickly gaining weight. These results were further supported by the lipid profiles of the mice: oil red O staining revealed that consumption of ARW reduced the accumulation of fat in the liver. Moreover, molecular data showed that supplementation with ARW induced the expression of the gene CYP7A1, which encodes cholesterol 7a-hydroxylase, the first and rate-limiting step in the bile acid synthetic pathway, the major site of regulation, and the primary mechanism for the removal of cholesterol from the body [123]. Based on these results, it appears that ARW exerts anti-obesity effects by inducing CYP7A1, which plays a critical role in cholesterol homeostasis in the body. Research in recent decades has highlighted the roof for ROS in health and disease. Generally, there is an association between the release of cytotoxic proteins, the production of ROS, inflammation, and the dominance of immune cytokines.

According to clinical studies involving ARW administered orally for senile disease treatment and the recuperation of hospitalized patients, there were neither distinct positive nor negative effects in aged patients [116]. However, all the measured blood parameters were within their normal ranges, including WBC, adiponectin, cholesterol, potassium, and liver enzymes associated with lipid metabolism [124]. Hence, it could be concluded that ARW does not induce adverse effects and might lead to a favorable body condition. According to our gathered studies, drinking ARW could lead to potentially beneficial health-promoting properties. Good water should be not only free of pollutants, but also have beneficial effects on health, such as the ability to detoxify the blood, strengthen the immune system, and counteract harmful free radicals. The trend of recent and existing articles on the effect of ARW on various lifestyle diseases proved to be significant, with data showing that there is a significant decrease in high-fat diet-induced obesity and total cholesterol, as well as improved lipid profiles of individuals. Taken together, alkaline water with a pH between 8 and 10 has positive feedback on diabetes and obesity, as well as a synergistic effect to lifestyle modification, such as exercise. It can reduce oxidative stress and inflammatory biomarkers, and scavenge and inhibit the production of ROS, thereby increasing antioxidant levels. It also improves hydration status and acid-base equilibrium, reducing the disturbances in the electrolyte content of the fluids and speeding up the rate of metabolism.

**Author Contributions:** Investigation, resources, data curation, writing—original draft, writing—review & editing, F.S.L.G.D.R., A.C.C.M., T.J.P.M., M.K.V.T., G.R.S., N.D.J., A.F., J.M.A., M.E.J.V.S.; Validation, F.S.L.G.D.R., A.F., J.M.A., M.E.J.V.S.; Visualization, A.C.C.M., A.F., J.M.A., M.E.J.V.S.; Conceptualization, supervision, M.E.J.V.S. All authors have read and agreed to the published version of the manuscript.
**Funding:** This study received no external funding.

**Institutional Review Board Statement:** Not applicable.

**Informed Consent Statement:** Not applicable.

**Data Availability Statement:** The data presented in this study are available within the article (tables and figures).

**Acknowledgments:** We would like to acknowledge and extend our gratefulness to the Department of Environmental Medical Biology, Wonju College of Medicine, Yonsei University, for sharing their expertise and allowing us to collaborate to write this manuscript.

**Conflicts of Interest:** The authors declare no conflict of interest.

## Abbreviations

| | |
|---|---|
| AEW | Alkaline Electrolyzed Water |
| AIW | Alkaline Induced Water |
| ARW | Alkaline Reduced Water |
| DM | Diabetes Mellitus |
| ERW | Electrolyzed Reduced Water |
| FA | Fatty Acid |
| FFA | Free Fatty Acid |
| FGF | Fibroblast Growth Factor |
| GPT | Glutamic Pyruvate Transaminase |
| GOT | Glutamic Oxaloacetate Transaminase |
| HDL | High-density Lipoproteins |
| HRW | Hydrogen-rich Water |
| IFN | Interferon |
| LDL | Low-density Lipoprotein |
| MRW | Mineral-induced Alkaline-reduced Water |
| NAFLD | Non-Alcoholic Fatty Liver Disease |

|       |                                       |
|-------|---------------------------------------|
| NASH  | Non-alcoholic Steatohepatitis          |
| NCD   | Non-communicable Diseases              |
| OLETF | Otsuka Long-Evans Tokushima Fatty      |
| ORP   | Oxidation-reduction Potential          |
| QoL   | Quality of Life                        |
| ROS   | Reactive Oxygen Species                |
| SCFAs | Short Chain Fatty Acids                |
| SOD   | Superoxide Dismutase                   |
| T1DM  | Type 1 Diabetes Mellitus               |
| T2DM  | Type 2 Diabetes Mellitus               |
| TLC   | Therapeutic Lifestyle Changes          |
| TG    | Triglycerides                          |
| TNF   | Tumor Necrosis Factor                  |
| VEGF  | Vascular Endothelial Growth Factor     |
| WHO   | World Health Organization              |

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
