# Peer review of "The Search for the Elixir of Life: On the Therapeutic Potential of Alkaline Reduced Water in Metabolic Syndromes"

_processes, doi:10.3390/pr9111876_

Round 1

Reviewer 1 Report

Dear authors,

The manuscript "The search for the elixir of Life: On the Therapeutic Potential of Alkaline Reduced Water in Metabolic Syndromes" is interesting and summarizes the association with the most important Non-communicable diseases and the interaction with the metabolic syndrome. There are some aspects that can be improved:

  • The introduction is too generic (the first 3 paragraphs) and should emphasize more the metabolic syndrome and the importance os ARW.
  • The authors should describe the inclusion and exclusion criteria used to select the papers and the databases used in the selection process.

Author Response

Point by Point Response:

From Reviewer 1

Comments and Suggestions: The manuscript “The search for the elixir of Life: On the Therapeutic Potential of Alkaline Reduced Water in Metabolic Syndromes” is interesting and summarizes the association with the most important non-communicable diseases and the interaction with the metabolic syndrome. There are some aspects that can be improved:

Comment (1): The introduction is too generic (the first three paragraphs) and should emphasize more the metabolic syndrome and the importance of ARW.

Response: We thank the reviewer for this input on the improvement of the introduction section. The authors think that the first three paragraphs are relevant to orient the readers from lifestyle diseases and metabolic syndrome. From there, mortality statistics and risk factors were stated to ground alkaline reduced water (ARW) relevant in this field. Moreover, as recommended we added additional lines in the introduction to emphasize the role of ARW as a potential treatment or therapy against metabolic syndrome. Necessary changes can be seen in line 89-109.

“To prevent the mentioned risk factors from manifesting, besides habit changes, pharmacological and therapeutical approaches are now also utilized.

In lieu of committing to a disciplined dietary regimen and physical activities, others have already resorted to medication as an effective gatekeeper to metabolic disorders. This is done primarily to those who have shown an inadequate change in their status even after administering preventive measures like exercise and a restricted diet [15]. Given the interconnectivity of the associated conditions under the umbrella term ‘metabolic syndrome,’ treating each at once remains a complex scientific enigma [16]. As such, experts recommended that physicians investigate these individually for now that no drug has yet addressed the syndrome holistically [17]. Although pharmacological approaches induce definitive and instantaneous effects in the prevention of each linked disease, therapeutic lifestyle changes (TLC) persist in being the core of clinical prophylaxis for metabolic syndrome, mainly due to the side effects imposed by drug intakes [18], making these less executable in the long-term. A cause célèbre is the introduction of alkaline ionized water (AIW) as it entails a propitious solution or, at the very least, provides a lens that might help in procuring a viable cure for metabolic syndrome”

Comment (2): The authors should describe the inclusion and exclusion criteria used to select the papers and the databases used in the selection process.

Response: We appreciate this valuable point. We have now added the inclusion and exclusion criteria used in the selection process of the papers and databases considered in line 402-410.

The inclusion and exclusion parameters that are used to select the Scopus-articles cited in Figure 3 and summarized in Table 1 revolved around the study design, date, and type of publication. Initially, screened articles using Scopus that focus on ARW or similar terms with relation to diseases associated with metabolic syndrome were filtered by excluding review articles and those studies with samples having pathogenic and genetically acquired diseases. Instead, articles that employed human clinical trials and in vitro experiments on study models involving animals, humans, and cell lines were chosen. To further filter, articles with a publication date before 2005 were removed from the selected list.

Check the attachment for the copy of the revised manuscript. Thank you.

Reviewer 2 Report

The authors collected several accounts from the journal articles pertaining to the effects of ARW consumption on metabolic syndrome and lifestyle diseases, specifically diabetes, obesity, NAFLD, and cancer.

This review article is timely, well conducted and very well written. I accept it in the current format.

Author Response

From Reviewer 2

Comments and Suggestions: The authors collected several accounts from the journal articles pertaining to the effects of ARW consumption on metabolic syndrome and lifestyle diseases, specifically diabetes, obesity, NAFLD, and cancer.

This review article is timely, well-conducted, and very well written. I accept it in the current format.

Response: Thank you very much for your insights and comments, very much appreciated.

Open Review Comment (3): English language and style are fine/minor spell check required.

Response: Grammar, English Language, and style were now improved and double-checked all throughout the manuscript. Thank you.

Kindly see attached file for the revised manuscript.
